# Soliton gas of the integrable Boussinesq equation and its generalised hydrodynamics

Thibault Bonnemain[1][*] and Benjamin Doyon[2]

**1** Department of Mathematics, King's College London, United Kingdom
**2** Univ. Lille, CNRS, UMR 8523 - PhLAM - Physique des Lasers Atomes et Molécules, F-59 000 Lille, France

[*] bonnemain.thibault@gmail.com

## Abstract

Generalised hydrodynamics (GHD) is a recent and powerful framework to study many-body integrable systems, quantum or classical, out of equilibrium. It has been applied to several models, from the delta Bose gas to the XXZ spin chain, the KdV soliton gas and many more. Yet it has only been applied to (1+1)-dimensional systems and generalisation to higher dimensions of space is non-trivial. We study the Boussinesq equation which, while generally considered to be less physically relevant than the KdV equation, is interesting as a stationary reduction of the (boosted) Kadomtsev-Petviashvili (KP) equation, a prototypical and universal example of a nonlinear integrable PDE in (2+1) dimensions. We follow a heuristic approach inspired by the Thermodynamic Bethe Ansatz in order to construct the GHD of the Boussinesq soliton gas. Such approach allows for a statistical mechanics interpretation of the Boussinesq soliton gas that comes naturally with the GHD picture. This is to be seen as a first step in the construction of the KP soliton gas, yielding insight on some classes of solutions from which we may be able to build an intuition on how to devise a more general theory. This also offers another perspective on the construction of anisotropic bidirectional soliton gases previously introduced phenomenologically by Congy et al (2021).

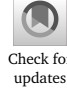

## Contents



# 1 Introduction

The theory of soliton gases, first introduced twenty years ago by Gennady El [1,2], has been the subject of rapidly growing interest in the past few years. This is in part due to the development of the notion of "integrable turbulence" coined and popularised by Zakharov [3], and to that of generalised hydrodynamics (GHD) independently introduced in both [4] and [5]. Those two approaches, though formally different, have a common aim which consists in investigating the statistical and emergent, hydrodynamical properties of large-scale, many-body, integrable systems out of equilibrium. As it happens, soliton gases are physically relevant mathematical objects that can be naturally interpreted both in terms of a soliton-dominated integrable turbulence and in terms of GHD.

The prime motivation behind the study of integrable turbulence and GHD lies in the fact that integrable systems are universal, arising from the asymptotic expansion of large classes of models, and are known to capture essential properties of several physical systems [6]. Moreover, a key aspect of integrable systems is that they feature an infinite number of conserved quantities (mass, momentum, energy, and their higher-spin versions) that constrain the dynamics and allow for exact solutions. Many techniques have been developed to that end, the most notable of which are perhaps the inverse scattering transform (IST) for classical field theories [7–9], and the Bethe Ansatz for quantum systems [10–12]. But the inherent complexity of many natural or experimentally observed phenomena, for instance due to initial or boundary conditions [13–15], often requires a statistical description which is beyond well-established mathematical techniques from the theory of integrable systems.

In particular strongly nonlinear random waves described by one-dimensional integrable systems, such as the Korteweg–de Vries (KdV) and the nonlinear Schrödinger (NLS) equations, have recently attracted significant attention from theoreticians [16–20] and experimentalists [21–27] alike. The theory of integrable turbulence then concerns itself with the statistical properties of the random wave field, for example in terms of probability density function, power spectrum and correlations, as discussed in Zakharov's seminal paper [3].

But those equations notably also feature solutions involving solitary waves called solitons that exhibit particle-like properties, such as elastic, pairwise interactions which only result in a simple position shift. Interacting solitons can then form large, complex and irregular statistical ensembles, first considered, again, by Zakharov [28]. This provides an alternative interpretation for the integrable turbulence of a soliton-dominated wave field in terms of a soliton gas. The focus is then put on the collective dynamics and kinetics of solitons, which are treated as interacting (quasi)particles characterised by their velocity (and/or amplitude) and the associated distribution function.

It is highly non-trivial that such a simple picture of a gas of solitons as a gas of particles should always hold, particularly for dense gases, as each soliton continuously overlaps and interacts with many others. But the structure of the kinetic equations for KdV and NLS soliton gases, rigorously derived in [1] and [17] from a thermodynamic-type limit of spectral finite gap solutions and their modulations [29], strongly supports this interpretation. This in turn allowed for some phenomenological constructions, notably in the case of bidirectionnal soliton gases [30], based on the so-called collision rate Ansatz. We refer the interested reader to the recent review [31] for an in depths discussion of soliton gas theory.

Given the aforementioned quasiparticle picture, the soliton gas approach is in many ways closer in spirit to GHD than it is to integrable turbulence. This correspondence was noted early [32, 33] and discussed at length in [34] in the context of the KdV equation. GHD was initially developed to tackle out-of-equilibrium, integrable, quantum, many-body systems by combining the (generalised) thermodynamic Bethe Ansatz (TBA) [35–37] and the hydrodynamic principles [38] to account for the local relaxation of fluid cells to generalised Gibbs ensembles (GGE) [39]. The need for such extensions comes from the fact that, in the case of integrable systems, the infinite number of conserved quantities is known to impede the process of "thermalisation" or, at least, it does in the traditional sense (e.g. there is no equipartition of energy). As it happens, integrable systems *do* thermalise, yet this notion needs to be generalised: entropy maximisation occurs, only it does with respect to *all* the infinitely many conserved quantities. This recent theory has quickly been applied to quantum gases, chains and field theories [40–42], providing exact results regarding their correlations and fluctuations [43–46]. It was then extended to describe classical integrable systems [47–55], to include higher order terms in the hydrodynamic expansion like diffusion [56–58] or dispersion [59], and to probe integrability breaking [60–64]. Notably, GHD and some of its extensions have been confirmed in experiments on cold atomic gases [65–67]. GHD is a recent but very active field of research and we refer the interested reader to the lecture notes and reviews [68–72].

This paper aims to investigate the soliton gas associated to the integrable Boussinesq equation

$$u_{tt} - u_{xx} = -\alpha \left[ 6 \left( u^2 \right)_{xx} + u_{xxxx} \right], \tag{1}$$

where $\alpha = \pm 1$. The $\alpha = 1$ case is typically referred to as the "good" Boussinesq equation, while $\alpha = -1$ corresponds to the "bad" one, a terminology that can be traced back to McKean [73].

It is important to stress that this equation is generally considered to be less physically relevant than the KdV equation [7]. Although it was historically derived as a model for propagation of waves in shallow water, it has since been noted that it is in fact not a valid model for shallow water hydrodynamics [74]. Moreover it has been argued that the only correct long shallow water wave model is the decoupled left and right running KdV equations [75], to which the Boussinesq equation reduces in the (relevant) small amplitude, weakly nonlinear regime [76, 77].

However, there are several meaningful incentives that motivate our study of the Boussinesq equation. From the point of view of physics, it arises naturally (for appropriate initial conditions) as the continuum limit of the Fermi-Pasta-Ulam-Tsingou chain [77, 78], and characterises stratified fluids at near (non-semisimple) double criticality [79]. It is arguably more

significant from the point of view of biology, as a simple generalisation of the Boussinesq equation – seemingly close to integrability, and still featuring stable solitons that interact almost elastically – has been proposed as a model for neural activity [80–82].

But perhaps the biggest incentive, going back to physics, comes from the fact that it can be seen as a stationary reduction of the (boosted) KP equation [7,83]

$$\left[u_t + 6(u^2)_x + u_{xxx}\right]_x + \alpha u_{yy} = 0. \tag{2}$$

The case $\alpha = 1$ is known as the KP2 equation, associated with the "good" Boussinesq equation, while $\alpha = -1$ corresponds to KP1 equation, associate to the "bad" one. The KP equation, in either version, has been used to model several physical phenomena such as the propagation of ion-acoustic waves in plasmas [84,85], of shallow water waves [86,87], or of a beam of light in nonlinear media [88,89]. It is a prototypical and universal example of a nonlinear integrable PDE in (2+1) dimensions and can be seen as a natural extension of the KdV equation to two spatial dimensions. As such, constructing the soliton gas or the GHD of the KP equation would be significant, as both theories have up to now been developed to deal with, and only applied to, (1 + 1)-dimensional systems, and generalisation to higher dimensions of space is non-trivial. We will undertake this endeavour in another paper [90]. The present discussion on the Boussinesq equation is then to be seen as a first step in the construction of the KP soliton gas, yielding insight on some classes of solutions from which we may be able to build an intuition on how to devise a more general theory.

In order to tackle the integrable Boussinesq equation, rather than the historical and more rigorous approach to soliton gas developed by El [1], we adopt here a more heuristic approach inspired by [34,48] and based on the TBA. This allows for a statistical mechanics interpretation of the Boussinesq soliton gas that comes naturally with the GHD picture, and thermodynamic quantities are then defined and evaluated (e.g. free energy, entropy, temperature). This also offers another perspective on the construction of anisotropic bidirectional soliton gases introduced phenomenologically in [30].

The paper is organised as follows. In Section 2 we review the Boussinesq equation and its main properties. In Section 3 we construct its thermodynamics: we write down integral equations for the exact free energy in generalised Gibbs ensembles, which take the form of the classical TBA. In Section 4 we phenomenologically construct its hydrodynamics: we derive its kinetic equation. Finally, we conclude in Section 5.

## 2 The integrable Boussinesq equation

The Boussinesq equation (1) describes the propagation of waves in (1+1) dimensional, weakly nonlinear, weakly dispersive systems, an example of which is shallow water [91, 92]. This is a bidirectional model: contrary to waves described by the KdV equation, those described by the Boussinesq equation can move in either direction. This last point will have important consequences on the GHD of the Boussinesq soliton gas as we shall see in Section 3. Finally, as mentioned in the previous section, depending on the sign of the parameter $\alpha$, the Boussinesq equation is referred to as "good" or "bad". We shall start our discussion by addressing this dichotomy.

### 2.1 The "good", the "bad" and the ill-posed

The "bad" Boussinesq equation is sometimes also called the ill-posed Boussinesq equation, because of its severe short-wave instability [93]. This can be made clear by linearising the

equation (1) about a constant background $u_0 > 0$, yielding the linear PDE

$$u_{tt} - u_{xx} = -\alpha \left[ 12 u_0 u_{xx} + u_{xxxx} \right]. \tag{3}$$

This equation admits solutions in the form $\exp[i(kx - \omega t)]$ with dispersion relation

$$\omega^2 = k^2 \left[ \alpha k^2 - 2\Omega_\alpha \right], \tag{4}$$

where $2\Omega_\alpha = 12\alpha u_0 - 1$. In the case of the "good" Boussinesq equation $\alpha = +1$, and one can see that perturbations of the constant solution are stable for $k^2 > \max[2\Omega_+, 0]$ and unstable otherwise, with, for $\Omega_+ > 0$, a maximal growth rate $i\omega = \Omega_+$ attained for $k = \sqrt{\Omega_+}$. However, if $\alpha = -1$, perturbations are unstable to all modes $k^2 > 2|\Omega_-|$, with unbounded growth rate, since $k$ is unbounded and $\omega$ grows monotonically with $k$. That means that the slightest perturbation of the initial data would instantly and utterly change the behaviour of the solution up to the smallest scales, and, as such, solutions of the "bad" Boussinesq equation with arbitrary initial conditions generically do not exist for positive time.

Fortunately, both for the "good" and "bad" Boussinesq equations, certain classes of solutions remain stable at all time. In particular, that is the case of the soliton solutions we will be considering in the remainder of this paper. For the sake of simplicity and legibility we will now drop the parameter $\alpha$ and focus exclusively on the "good" Boussinesq equation. All the upcoming results can be straightforwardly extended to the "bad" Boussinesq equation as is discussed in Appendix A.

## 2.2 Integrability and conserved quantities

In 1973, Zakharov showed that the Boussinesq equation (1) was integrable, and solvable via IST [94], by introducing the Lax pair

$$\begin{cases} \hat{L} = i \left[ \dfrac{d^3}{dx^3} + u \dfrac{d}{dx} + \dfrac{d}{dx} u \right] - \sqrt{\dfrac{4}{3}}\, w_x, \\[4mm] \hat{A} = \sqrt{\dfrac{3}{4}} \dfrac{d^2}{dx^2} + \sqrt{\dfrac{4}{3}}\, u, \end{cases} \tag{5}$$

where the function $w(x,t)$ is defined by $u_t = w_{xx}$. Indeed, one can see by direct substitution that the Lax equation $\hat{L}_t = i[\hat{L}, \hat{A}]$ is equivalent to the original equation (1) under the so-called *isospectrality condition*, i.e. provided the spectrum of the operator $\hat{L}$ is invariant.

In the same paper, Zakharov also devised a recurrence formula to write an infinite set of integrals of motion in involution. We shall not discuss this recurrence of formula or how it came to be, however, as an illustration, we provide here the expressions of the first four conserved charges (densities):

$$\begin{aligned} Q_1 &= \int_{\mathbb{R}} dx\, u, \\[2mm] Q_2 &= \int_{\mathbb{R}} dx\, w_x, \\[2mm] Q_3 &= \int_{\mathbb{R}} dx\, u w_x, \\[2mm] Q_4 &= \int_{\mathbb{R}} dx \left[ \frac{u^2}{2} + \frac{(w_x)^2}{2} - 2u^3 + \frac{(u_x)^2}{2} \right], \end{aligned} \tag{6}$$

which all commute with respect to the canonical Poisson bracket defined as the bilinear map

$$\{F; G\} = \int_{\mathbb{R}} dx \left( \frac{\delta F}{\delta u(x)} \frac{\delta G}{\delta w(x)} - \frac{\delta F}{\delta w(x)} \frac{\delta G}{\delta u(x)} \right), \tag{7}$$

for real-analytic functionals in the sense of [9]. In particular, the fourth charge $Q_4$ plays the role of a Hamiltonian, and one can alternatively write the Boussinesq equation (1) in the Hamiltonian form

$$\begin{cases} u_t = w_{xx} = \{Q_4; u(x)\} = -\dfrac{\delta Q_4}{\delta w}, \\[2mm] w_t = u - 6u^2 - u_{xx} = \{Q_4; w(x)\} = \dfrac{\delta Q_4}{\delta u}. \end{cases} \tag{8}$$

Finally, given the definition of $w$, the current associated to the conserved density $u$ (for $Q_1$) is $-w_x$, but we see that $w_x$ is also the conserved density for $Q_2$. Thus $w_x$ is a self-conserved current, in particular by defining

$$J_1 \equiv \int_{\mathbb{R}} dt \, w_x, \tag{9}$$

one has

$$\partial_x J_1 = \partial_t J_1 = 0, \tag{10}$$

if we assume that $u$, $w$ and their derivatives vanish as $|t| \to \infty$ and/or $|x| \to \infty$, which we will be doing since we are only interested in reflectionless (i.e., pure soliton) solutions [76]. The presence of a self-conserved current justifies the use of the collision rate Ansatz further down the line [50, 95].

## 2.3 Soliton solutions

Although very powerful, the IST formalism is not best suited for what we aim to do here. It will instead be more convenient to use Hirota's bilinear method [76] to construct the soliton solutions and extract their 2-body scattering shift. Indeed, Hirota showed in 1973 that

$$u(x, t) = \log[\tau(x, t)]_{xx}, \tag{11}$$

is a solution of the Boussinesq equation (1) provided the $\tau-$function solves

$$\left[ (\partial_t - \partial_{t'})^2 - (\partial_x - \partial_{x'})^2 - (\partial_x - \partial_{x'})^4 \right] \tau(x, t) \tau(x', t') = 0. \tag{12}$$

Notably, $N-$soliton solutions are obtained from $\tau-$functions of the form

$$\tau(x, t) = 1 + \sum_{n=1}^{N} \sum_{{}_N C_n} a(i_1, i_2, \cdots, i_n) \exp(\theta_{i_1}(x, t) + \theta_{i_2}(x, t) + \cdots + \theta_{i_n}(x, t)), \tag{13}$$

where ${}_N C_n$ indicates summation over all combinations of $n$ elements taken from $N$. Both the weights $a(i_1, i_2, \cdots, i_n)$ and the arguments of the exponential $\theta_i$ in (13) need to be discussed: they can be made explicit by simply plugging the Ansatz (13) in Eq. (12) and can be interpreted in terms of the properties of the solitons.

An $N-$soliton solution is parameterised by three sets of $N$ parameters: $0 < \eta_i < 1$, $\epsilon_i = \pm 1$ and $x_i^0 \in \mathbb{R}$, for $i = 1, \ldots, N$. The function $\theta_i(x, t)$ can be seen as the "phase" associated with a soliton indexed by the spectral doublet[1] $(\eta_i, \epsilon_i)$ and an "initial position" $x_i^0$

$$\theta_i(x, t) = \eta_i \left( x - \epsilon_i t \sqrt{1 - \eta_i^2} - x_i^0 \right), \tag{14}$$

---

[1]The names "spectral doublet" or "spectral parameters" come from the fact those parameters can directly be linked to the discrete eigenvalues of the Lax operator $\hat{L}$ defined in Eq. (5).

where $\epsilon_i = +1, -1$ distinguishes between right- and left-moving solitons, respectively, which have speeds $v_i = \sqrt{1-\eta_i^2}$ and velocities (that is, including the direction) $v_i = \epsilon_i v_i$. Additionally, the coefficient $a(i_1, i_2, \cdots, i_n)$, in front of the exponential in Eq. (13), can be expressed in terms of the 2-body phase shifts $\varphi_{ij}$

$$a(i_1, i_2, \cdots, i_n) = \prod_{k<l}^{n} \exp \varphi_{i_k i_l}, \tag{15}$$

where

$$\varphi_{ij} = \log \frac{\left(\epsilon_i \sqrt{1-\eta_i^2} - \epsilon_j \sqrt{1-\eta_j^2}\right)^2 - 3(\eta_i - \eta_j)^2}{\left(\epsilon_i \sqrt{1-\eta_i^2} - \epsilon_j \sqrt{1-\eta_j^2}\right)^2 - 3(\eta_i + \eta_j)^2}. \tag{16}$$

Note that for small values of $\eta_i$ and $\eta_j$, we recover the KdV phase shift in the case of overtaking collisions

$$\varphi_{ij}^{\text{KdV}} = 2\log \left| \frac{\eta_i - \eta_j}{\eta_i + \eta_j} \right|, \tag{17}$$

and we shall discuss the link between KdV and Boussinesq in more details at the end of Section 4.2.

## 2.4 Illustration: One- and two-soliton solutions

To ensure these interpretations are correct, let us first look at the 1-soliton solution

$$\begin{aligned} u_1(x,t) &= \log\left[1 + e^{\theta_1}\right]_{xx} \\ &= \left(\frac{\eta_1}{2}\right)^2 \text{sech}^2\left[\frac{\eta_1}{2}\left(x - \epsilon_1 t \sqrt{1-\eta_1^2} - x_1^0\right)\right], \end{aligned} \tag{18}$$

which confirms $\theta_1$ can be seen as a phase. Note how, contrary to the phenomenology of the KdV equation, the speed of the soliton $v_i = \sqrt{1-\eta_i^2}$ decreases as its amplitude grows. Given the 1-soliton solution (18), we can define the set of $\{h_n\}$'s, namely the amount of charge $Q_n$ carried by a single soliton of spectral doublet $(\eta, \epsilon)$. Here are the first four, up to a multiplicative constant (which will be irrelevant for us), given definitions (6)

$$\begin{aligned} h_1(\eta, \epsilon) &\propto \eta, \\ h_2(\eta, \epsilon) &\propto \epsilon\eta\sqrt{1-\eta^2}, \\ h_3(\eta, \epsilon) &\propto \epsilon\eta^3\sqrt{1-\eta^2}, \\ h_4(\eta, \epsilon) &\propto 5\eta^3 - 4\eta^5. \end{aligned} \tag{19}$$

Now, let us focus on the 2-soliton solution

$$u_2(x,t) = \log\left[1 + e^{\theta_1} + e^{\theta_2} + a_{12}e^{\theta_1+\theta_2}\right]_{xx}. \tag{20}$$

As an example assuming that $\eta_1 < \eta_2$, $\epsilon_1 = +1$ and $\epsilon_2 = \pm1$, so that $v_1 > v_2$, let us look in the vicinity of $x \approx v_1 t$. If we take the limit $t \to -\infty$, so that $\theta_1$ stays finite but $\theta_2 \to -\infty$, we directly recover the one soliton solution (18)

$$u_2(x,t) \approx u_1(x,t). \tag{21}$$

However, still looking in the vicinity of $x \approx v_1 t$, if this time we take the limit $t \to \infty$, then $\theta_2 \to \infty$ and we obtain

$$\begin{aligned} u_2(x,t) &\approx \log\left[e^{\theta_2}(1 + a_{12}e^{\theta_1})\right]_{xx} \\ &= \left(\frac{\eta_1}{2}\right)^2 \text{sech}^2\left[\frac{\eta_1}{2}\left(x - \epsilon_1 t \sqrt{1-\eta_1^2} - x_1^0\right) + \frac{\varphi_{12}}{2}\right], \end{aligned} \tag{22}$$

showing that $\varphi_{ij}/\eta_i$ indeed plays the role of the phase shift of soliton 1 as it crosses soliton 2. Here, soliton 1 is right-moving, $\epsilon_1 = +$, and we have considered simultaneously the overtaking, $\epsilon_2 = +$, and the head-on, $\epsilon_2 = -$, collisions.

## 2.5 Regularity conditions for the $N-$soliton solution

In full generality, given a $\tau-$function of form (13), we may asymptotically write the $N-$soliton solution as

$$u_N(x,t) \approx \sum_{i=1}^{N} \left(\frac{\eta_i}{2}\right)^2 \mathrm{sech}^2\left[\frac{\eta_i}{2}\left(x - \epsilon_i t \sqrt{1-\eta_i^2} - x_i^{\pm}\right)\right], \quad \text{as} \quad t \to \pm\infty, \qquad (23)$$

where $x_i^-$ and $x_i^+$ are respectively called the "in" and "out" impact parameters, and are related by

$$x_i^+ - x_i^- = \frac{1}{\eta_i}\sum_{j\neq i}\varphi_{ij}. \qquad (24)$$

In effect, $N-$soliton solutions are entirely characterised by the set of triplets $\{\eta_i;\, x_i^-;\, \epsilon_i\}_{i=1}^{N}$.

Let us now consider a $N-$soliton solution, $M$ of which are left-moving and $(N-M)$ of which are right-moving. By convention, we shall order the spectral parameters according to the velocities of the solitons $v_i = \epsilon_i\sqrt{1-\eta_i^2}$, such that

$$v_1 < \cdots < v_M < 0 < v_{M+1} < \cdots < v_N, \qquad (25)$$

or, equivalently

$$\eta_1 < \cdots < \eta_M, \quad \text{and} \quad \eta_N < \cdots < \eta_{M+1}. \qquad (26)$$

Without loss of generality, because of Galilean invariance, we can assume that the slowest (largest) soliton is right-moving, $\eta_{M+1} > \eta_M$.

Noting that the expression of the 2-body phase shift (16), in terms of the spectral parameters of the interacting solitons, only depends on the product $\epsilon_i\epsilon_j = \pm 1$, and not on the direction in which they are moving, we define the overtaking position shift

$$\Delta_{\mathrm{O}}(\eta_i,\eta_j) \equiv \mathrm{sgn}(v_j-v_i)\frac{\varphi_{\mathrm{O}}(\eta_i,\eta_j)}{\eta_i} = \frac{\mathrm{sgn}(v_j-v_i)}{\eta_i}\log\frac{\left(\sqrt{1-\eta_i^2}-\sqrt{1-\eta_j^2}\right)^2 - 3(\eta_i-\eta_j)^2}{\left(\sqrt{1-\eta_i^2}-\sqrt{1-\eta_j^2}\right)^2 - 3(\eta_i+\eta_j)^2}, \qquad (27)$$

and the head-on position shift

$$\Delta_{\mathrm{H}}(\eta_i,\eta_j) \equiv -\frac{\varphi_{\mathrm{H}}(\eta_i,\eta_j)}{\epsilon_i\eta_i} = \frac{-1}{\epsilon_i\eta_i}\log\frac{\left(\sqrt{1-\eta_i^2}+\sqrt{1-\eta_j^2}\right)^2 - 3(\eta_i-\eta_j)^2}{\left(\sqrt{1-\eta_i^2}+\sqrt{1-\eta_j^2}\right)^2 - 3(\eta_i+\eta_j)^2}. \qquad (28)$$

An important aspect of the system we are considering is that the argument of the log in the position shifts (27) and (28) is not always positive for any couple $(\eta_i,\eta_j)$. This means the $N-$soliton solution does not necessarily remain regular for all time. Given the ordering (26), the $N-$soliton solution is always regular for either [96]

$$\begin{cases} \eta_{M+1} < \dfrac{\sqrt{3}}{2}, \\[2mm] \eta_M < \dfrac{1}{2}\left|\eta_{M+1} - \sqrt{3\left(1-\eta_{M+1}^2\right)}\right|, \end{cases} \quad \text{or} \quad \begin{cases} \dfrac{\sqrt{3}}{2} < \eta_{M+1} < 1, \\[2mm] \eta_{M+2} < \dfrac{1}{2}\left[\eta_{M+1} + \sqrt{3\left(1-\eta_{M+1}^2\right)}\right], \\[2mm] \eta_M < \dfrac{1}{2}\left[\eta_{M+1} - \sqrt{3\left(1-\eta_{M+1}^2\right)}\right], \end{cases} \qquad (29)$$

if $\eta_{M+2} > \eta_M$, or

$$\begin{cases} \eta_{M+1} < \dfrac{\sqrt{3}}{2} \,, \\[2mm] \eta_{M+2} < \dfrac{1}{2} \left| \eta_{M+1} - \sqrt{3\left(1-\eta_{M+1}^2\right)} \right| , \end{cases} \quad \text{or} \quad \begin{cases} \dfrac{\sqrt{3}}{2} < \eta_{M+1} < 1 \,, \\[2mm] \eta_M < \dfrac{1}{2}\left[ \eta_{M+1} - \sqrt{3\left(1-\eta_{M+1}^2\right)} \right] , \end{cases} \tag{30}$$

if $\eta_M > \eta_{M+2}$. Recall that, because of ordering (25) and (26), the highest value of the spectral parameters is $\eta_{M+1}$ followed either by $\eta_M$ or $\eta_{M+2}$.[2] Note that allowing for complex phases $\theta_i$, a complex phase shift would result in a soliton transforming into an antisoliton and vice-versa [97], however this is not something we will be considering here.

From the expressions of the position shifts (27) and (28), one may see that solitons with greater velocity receive positive overtaking shifts,[3] while solitons with smaller velocities receive negative shifts. In other words, smaller solitons are shifted forward with respect to their direction of propagation, and larger solitons are shifted backward as a result of overtaking interactions. In case of head-on collisions, solitons are shifted in the same direction, regardless of their amplitude: if $\varphi_{\mathrm{H}} > 0$ right-moving solitons are shifted to the left and left-moving to the right; if $\varphi_{\mathrm{H}} < 0$ right-movers are shifted to the right and left-movers to the left. While the position shift induced by overtaking collisions can always be interpreted as solitons repelling each other, $\varphi_{\mathrm{O}} < 0$, head-on interactions can be seen as attractive for $\varphi_{\mathrm{H}} > 0$ and bound states may appear.

## 2.6 Bound states

On top of all this, in [96] Lambert et al. identified what they call degenerate and resonant soliton solutions of the Boussinesq equation (1), which are related to the Miles resonant solutions of the KP equation [98–100]. Those correspond to regular solutions for which the phase shift between two solitons may become infinite, $\varphi_{ij} \to -\infty$ or $\varphi_{ij} \to \infty$ respectively for what they call degenerate or resonant solutions. Those can only occur under specific conditions and, given the ordering (25) and (26), one can identify three cases:

- Overtaking degeneracy for

$$\begin{cases} \dfrac{\sqrt{3}}{2} < \eta_{M+1} < 1 \,, \\[2mm] \eta_{M+2} = \dfrac{1}{2}\left[ \eta_{M+1} + \sqrt{3(1-\eta_{M+1}^2)} \right] , \\[2mm] \eta_M < \dfrac{1}{2}\left[ \eta_{M+1} - \sqrt{3(1-\eta_{M+1}^2)} \right] . \end{cases} \tag{31}$$

- Head-on degeneracy for

$$\begin{cases} \dfrac{\sqrt{3}}{2} < \eta_{M+1} < 1 \,, \\[2mm] \eta_M = \dfrac{1}{2}\left[ \eta_{M+1} - \sqrt{3(1-\eta_{M+1}^2)} \right] , \\[2mm] \eta_{M+2} < \dfrac{1}{2}\left[ \eta_{M+1} + \sqrt{3(1-\eta_{M+1}^2)} \right] . \end{cases} \tag{32}$$

- Head-on resonance for

$$\begin{cases} \dfrac{1}{2} < \eta_{M+1} < \dfrac{\sqrt{3}}{2} \,, \\[2mm] \eta_{M-1} = \dfrac{1}{2}\left[ \eta_{M+1} - \sqrt{3(1-\eta_{M+1}^2)} \right] . \end{cases} \tag{33}$$

---

[2]Note that this is slightly different from the description in [96].
[3]Note that $\varphi_{\mathrm{O}}(\eta_i, \eta_j) < 0 \ \forall \ (i, j)$ given the constraint (29).

Phenomenologically those three cases are similar and, in fact, any one can be made into another through time and/or space reversal symmetry. They correspond to inelastic processes: a soliton of large amplitude $\eta_{M+1}$ decays in two smaller ones, of amplitudes $\eta_M$ and $(\eta_{M+1} - \eta_M)$, moving in opposite direction (head-on degeneracy); a right-moving soliton of large amplitude $(\eta_{M+1} + \eta_M)$ decays in two smaller ones, $\eta_{M+1}$ and $\eta_M$ (head-on resonance); two smaller solitons, $\eta_{M+2}$ and $(\eta_{M+1} - \eta_{M+2})$, moving in opposite direction merge into a larger one $\eta_{M+1}$ (overtaking degeneracy). Those degenerate/resonant solutions asymptotically appear either as $(N-1)$−soliton solutions (at $t \to -\infty$ for head-on degeneracy or resonance, at $t \to \infty$ for overtaking degeneracy) or as $N$−soliton solutions (at $t \to \infty$ for head-on degeneracy or resonance, at $t \to -\infty$ for overtaking degeneracy), while the other $(N-2)$ "spectator" solitons remain unaffected. Lastly, one may consider a combination of the two degenerate cases in which $\eta_{M+2} = \frac{1}{2}\left[\eta_{M+1} + \sqrt{3(1 - \eta_{M+1}^2)}\right]$ and $\eta_M = \frac{1}{2}\left[\eta_{M+1} - \sqrt{3(1 - \eta_{M+1}^2)}\right]$ (which implies $\eta_{M+1} = \eta_{M+2} + \eta_M$) appearing as a degenerate $(N-1)$−soliton solution with $(N-3)$ spectator solitons.

Because resonances happen at the boundaries of the regularity region, only the cases discussed in this section should remain regular at all times, which has been confirmed in [101], and no four soliton processes are possible (contrary to KP which feature resonant tetrads [96]). As such this is not an effect that we expect to be thermodynamically relevant.

# 3 Thermodynamics

As we have seen in the previous section, $N$−soliton solutions of the Boussinesq equation (1) are entirely specified by the triplets $\{\eta_i; \ x_i^-; \ \epsilon_i\}_{i=1}^N$ characterising each individual soliton. This, along with the asymptotic form (23), naturally suggests an interpretation of solitons in terms of quasi-particles, and of a *soliton gas* as a literal gas of solitons. Naturally, this interpretation is only to be taken as a useful tool for reasoning but not as a rigorous description: solitons are completely unrecognisable once in the gas and constantly overlap with one another. Nevertheless, one expects that it be possible to determine the soliton content of any large enough region of space by a "time-of-flight" thought experiment: one extracts the Boussisnesq field from this region and puts it in the vacuum (i.e. the field is to zero away from the chosen region), and then lets the full, new field configuration on the line evolve in time until separate solitons are seen. In this way, one may determine the approximate spatially resolved soliton content on the line, and, it turns out, a heuristic argument taking its root in the "gas of solitons" interpretation is rather accurate to predict both its dynamics and fluctuations. In this section, from such an heuristic argument, we shall develop the thermodynamics of the Boussinesq soliton gas characterised by the associated Generalised Gibbs Ensemble (GGE). Not only will we derive the bidirectional kinetic equations phenomenologically proposed in [30], but we will also describe the gas in the language of statistical mechanics, introducing an entropy, free energy, temperature, etc.

## 3.1 Classical TBA

Following the heuristic procedure previously applied to the classical Toda system in [48] and to the KdV equation in [34], we may construct the classical TBA of the Boussinesq soliton gas, given either condition (29) or (30) are satisfied. To that end, we shall consider solitons as quasi-particles indexed by $i$, of position $x_i^t$ at time $t$, of positive or negative velocity, and spectral parameter $\eta_i$. We will assume that, at all times, the only effect of the interactions between solitons is the shift in position introduced in Section 2.5. This is motivated by the fact

that solitons asymptotically behave as free particles

$$x_i^t = x_i^\pm + v_i t, \quad \text{for} \quad t \to \pm\infty, \tag{34}$$

that are spatially ordered according to their velocity.

Now, let us consider a $N-$soliton solution initially supported on a finite interval $[0, L]$ in the sense that $u_N(x, t = 0)$ decays exponentially fast if $x \notin [0, L]$ or, in terms of the quasi-particle picture $\{x_i^0 \in [0, L], \forall i = 1, 2 \cdots, N\}$. Recall that we take the labelling ordered as per velocities, (25). If we assume that the right-moving quasi-particle $i$ is initially the left-most particle, $x_i^0 = 0$, then, between $t \to -\infty$ and $t = 0$, it almost surely only incurred shifts from faster right-moving solitons, such that we can evaluate its (in) impact parameter $x_i^{\text{left}}$ as

$$0 = x_i^{\text{left}} + \frac{1}{\eta_i} \sum_{j=i+1}^{N} \log \frac{\left(\sqrt{1-\eta_i^2} - \sqrt{1-\eta_j^2}\right)^2 - 3(\eta_i - \eta_j)^2}{\left(\sqrt{1-\eta_i^2} - \sqrt{1-\eta_j^2}\right)^2 - 3(\eta_i + \eta_j)^2}. \tag{35}$$

Similarly, if we assume that the right-moving quasi-particle $i$ is initially the right-most particle, $x_i^0 = L$, then, between $t \to -\infty$ and $t = 0$, it almost surely only incurred shifts from slower right-moving solitons and from all left-moving ones, and its impact parameter $x_i^{\text{right}}$ takes the form

$$L = x_i^{\text{right}} - \frac{1}{\eta_i} \left[ \sum_{j=M+1}^{i-1} \log \frac{\left(\sqrt{1-\eta_i^2} - \sqrt{1-\eta_j^2}\right)^2 - 3(\eta_i - \eta_j)^2}{\left(\sqrt{1-\eta_i^2} - \sqrt{1-\eta_j^2}\right)^2 - 3(\eta_i + \eta_j)^2} \right.$$
$$\left. + \sum_{j=1}^{M} \log \frac{\left(\sqrt{1-\eta_i^2} + \sqrt{1-\eta_j^2}\right)^2 - 3(\eta_i - \eta_j)^2}{\left(\sqrt{1-\eta_i^2} + \sqrt{1-\eta_j^2}\right)^2 - 3(\eta_i + \eta_j)^2} \right]. \tag{36}$$

In other words, if a quasiparticle $i$ is initially located in a box $[0, L]$, its impact parameter must satisfy $x_i^- \in [x_i^{\text{left}}, x_i^{\text{right}}]$. By subtracting Eq (35) to Eq (36) we may compute the length of the asymptotic space for right-movers $L_i^{\text{r}} = x_i^{\text{right}} - x_i^{\text{left}}$

$$L_i^{\text{r}} = L + \frac{1}{\eta_i} \left[ \sum_{j=M+1, \, j\neq i}^{N} \log \frac{\left(\sqrt{1-\eta_i^2} - \sqrt{1-\eta_j^2}\right)^2 - 3(\eta_i - \eta_j)^2}{\left(\sqrt{1-\eta_i^2} - \sqrt{1-\eta_j^2}\right)^2 - 3(\eta_i + \eta_j)^2} \right.$$
$$\left. + \sum_{j=1}^{M} \log \frac{\left(\sqrt{1-\eta_i^2} + \sqrt{1-\eta_j^2}\right)^2 - 3(\eta_i - \eta_j)^2}{\left(\sqrt{1-\eta_i^2} + \sqrt{1-\eta_j^2}\right)^2 - 3(\eta_i + \eta_j)^2} \right]. \tag{37}$$

A similar reasoning can be carried out for left-moving solitons and one may find that

$$L_i^{\text{l}} = L + \frac{1}{\eta_i} \left[ \sum_{j=1, \, j\neq i}^{M} \varphi_{\text{O}}(\eta_i, \eta_j) + \sum_{j=M+1}^{N} \varphi_{\text{H}}(\eta_i, \eta_j) \right]. \tag{38}$$

Introducing two functions $\mathcal{K}_{N,M}^{\text{r}}(\eta)$ and $\mathcal{K}_{N,M}^{\text{l}}(\eta)$ that respectively interpolate between the $(L_i^{\text{r}}/L)$'s and the $(L_i^{\text{l}}/L)$'s for fixed couples of $(N, M)$, we may then define the right- and left-asymptotic space densities, $\mathcal{K}_{\text{r}}(\eta)$ and $\mathcal{K}_{\text{l}}(\eta)$, by taking the thermodynamic limit: $L \to \infty$, $N \to \infty$, $M \to \infty$, while keeping the ratios $N/L = \varkappa$ and $M/N = \gamma$ finite. Then, from

Eqs (37) and (38) one may write

$$
\begin{cases}
\mathcal{K}_{\mathrm{r}}(\eta) = 1 + \dfrac{1}{\eta}\left[\displaystyle\int_{\Gamma_{\mathrm{r}}} \mathrm{d}\mu\ \varphi_{\mathrm{O}}(\eta,\mu)\rho_{\mathrm{r}}(\mu) + \int_{\Gamma_{\mathrm{l}}} \mathrm{d}\mu\ \varphi_{\mathrm{H}}(\eta,\mu)\rho_{\mathrm{l}}(\mu)\right], \\[3mm]
\mathcal{K}_{\mathrm{l}}(\eta) = 1 + \dfrac{1}{\eta}\left[\displaystyle\int_{\Gamma_{\mathrm{l}}} \mathrm{d}\mu\ \varphi_{\mathrm{O}}(\eta,\mu)\rho_{\mathrm{l}}(\mu) + \int_{\Gamma_{\mathrm{r}}} \mathrm{d}\mu\ \varphi_{\mathrm{H}}(\eta,\mu)\rho_{\mathrm{r}}(\mu)\right],
\end{cases}
\tag{39}
$$

where $\Gamma_{\mathrm{r}} \subset [0,1]$ and $\Gamma_{\mathrm{l}} \subset [0,1]$ represent the spectral support of right- and left-moving solitons, respectively, and where

$$
\begin{cases}
\rho_{\mathrm{r}}(\eta) = \dfrac{\varkappa(1-\gamma)}{N-M} \displaystyle\sum_{i=M+1}^{N} \delta(\eta-\eta_i), \\[3mm]
\rho_{\mathrm{l}}(\eta) = \dfrac{\varkappa\gamma}{M} \displaystyle\sum_{i=1}^{M} \delta(\eta-\eta_i),
\end{cases}
\tag{40}
$$

are the right- and left-densities of states (DOS) expressed in terms of the empirical densities of spectral points and of the spatial densities of solitons. Equations (39) are akin to the nonlinear dispersion relations (NDR's) of soliton gases (see [31] for a comprehensive review on the topic), this time for two types of quasi-particles, in which

$$
\sigma_{\mathrm{l}}(\eta) \equiv \frac{\eta\mathcal{K}_{\mathrm{l}}(\eta)}{\rho_{\mathrm{l}}(\eta)}, \quad \text{and} \quad \sigma_{\mathrm{r}}(\eta) \equiv \frac{\eta\mathcal{K}_{\mathrm{r}}(\eta)}{\rho_{\mathrm{r}}(\eta)},
\tag{41}
$$

would play the role of the left- and right- spectral scaling functions [31].

Before we continue, some remarks are in order. First, one should note that, in effect, left- and right- moving solitons are to be considered as two different particle types with their distinct DOS's. That is because the shifts for head-on and overtaking collision are different and one must keep track of each type of collisions. Second, echoing the discussion carried out in Section 4.5 of [34] regarding the ambiguity associated with the choice of the GHD momentum function, we should clarify which convention we shall adopt in the rest of the paper.

## 3.2   Conventional choice of the momentum function

As argued in [34] the parametrisation of solitons as quasi-particles in terms of the spectral and directional parameters $\eta$ and $\epsilon$ is somewhat arbitrary. When defining the momentum function $P(\eta,\epsilon)$ of the quasi-particles associated with solitons in our GHD construction, there are, a priori, two natural choices: the physical momentum of the soliton

$$
P_{\mathrm{phys}}(\eta,\epsilon) \equiv h_2(\eta,\epsilon) = \epsilon\eta\sqrt{1-\eta^2},
\tag{42}
$$

or its velocity

$$
P_{\mathrm{vel}}(\eta,\epsilon) \equiv v(\eta,\epsilon) = \epsilon\sqrt{1-\eta^2}.
\tag{43}
$$

For the purpose of constructing the GHD of the Boussinesq soliton gas, we could use the same convention as [34] and express all the upcoming results in terms of the momentum of the quasi-particle $v$, and not in terms of the "physical" momentum of the soliton $h_2$. However there is a third alternative which is less physically meaningful but formally more practical.

This comes from the fact that GHD was originally developed to study quantum integrable systems, in which the momentum function $P_{\mathrm{quant}}$ is unambiguous. In these systems the scattering phase $\varphi_{\mathrm{quant}}$ is related to the scattering shift $\Delta_{\mathrm{quant}}$ by the relation

$$
\varphi_{\mathrm{quant}}(\eta) = P'_{\mathrm{quant}}(\eta)\Delta_{\mathrm{quant}}(\eta),
\tag{44}
$$

and comparing with Eqs. (27)-(28), we construct

$$P(\eta, \epsilon) \equiv P_{\text{GHD}}(\eta, \epsilon) = \epsilon \frac{\eta^2}{2}. \tag{45}$$

In the upcoming sections, this parametrisation is the one that will yield the most compact and easiest to handle expressions. In effect, when it comes to the KdV equation discussed in [34], the conventions (43) and (45) are equivalent up to a multiplication by 8. The fact that the Boussinesq equation actually differentiates between these two conventions extends the previous discussion of [34] and provides a better way of selecting the arbitrary momentum function through Eq (44).

### 3.3 Partition function

In this section we will build up on the previous TBA-inspired approach to construct the thermodynamics of our soliton gas and the associated GGE. This can be done by writing the generalised Gibbs measure for an ensemble of $N$−soliton solutions and the corresponding partition function which, on a formal level, takes the form

$$\mathcal{Z}_N = \int \mathcal{D}[u_N] \exp\left(S[u_N] - W[u_N]\right), \tag{46}$$

where $S$ is the entropy of the gas to be defined, and $W$ a Lagrange parameter, the Gibbs weight, to account for the infinite set of conservation laws

$$W = \sum_{n=1}^{\infty} \beta_n Q_n, \tag{47}$$

associated with the infinite set of inverse temperatures $\{\beta_n\}_{n=1}^{\infty}$. At this point, expression (46) is admittedly rather unwieldy. However, since the $N$−soliton solution is entirely specified by the triplets $\{\eta_i; \ x_i^-; \ \epsilon_i\}_{i=1}^N$ (with $\epsilon_i = -1$ for $i = 1, 2, \ldots M$, and $\epsilon_i = 1$ for $i = M+1, M+2, \ldots N$), one may instead write its partition function in terms of the asymptotic coordinates

$$\mathcal{Z}_N = \sum_{M=0}^{N-1} \frac{M!(N-M)!}{(N!)^2} \int_{\Gamma_l^M \times \Gamma_r^{N-M} \times \mathbb{R}^N} \prod_{i=1}^{N} \frac{\mathrm{d}P(\eta_i)}{2\pi} \mathrm{d}x_i^-$$
$$\times \exp\left[-\sum_{i=1}^{N} w(\eta_i, \epsilon_i)\right] \chi\left(u_N(x, t=0) < \varepsilon_x, \ x \notin [0, L]\right), \tag{48}$$

where $\varepsilon_x \to 0$ fast enough as $\max(-x, x-L) \to \infty$ and

$$\mathrm{d}P(\eta_i) = \eta_i \mathrm{d}\eta_i. \tag{49}$$

This amounts to counting all possible realisations of a $N$-soliton solution $u_N$, with solitonic Gibbs weights $w$, and given the constraint that, at $t = 0$, it is supported on a finite interval $[0, L]$, represented by the indicator function $\chi$. Note that, in summing/integrating over the full spectral space of each solitons $(\eta_i, \epsilon_i)$, we only explicitly integrate over the $\eta_i$'s, as the sum over the $\epsilon_i$'s is already taking into account by the combinatoric factor. The solitonic weights $w(\eta, \epsilon)$ are defined as the individual contribution of a single soliton to the weight $W$ so that

$$w(\eta, \epsilon) \equiv \sum_{n} \beta_n h_n(\eta, \epsilon), \qquad W = \sum_{i=1}^{N} w(\eta_i, \epsilon_i). \tag{50}$$

An important point is that there is in fact no need to write $w(\eta, \epsilon)$ as an expansion in the functions $h_n(\eta, \epsilon)$ – any function $w(\eta, \epsilon)$ which is bounded from below will do. Of course, it is not clear that an arbitrary function of $\eta, \epsilon$ can be written as an expansion of the $h_n(\eta, \epsilon)$'s, whose first few are written in (19). In view of Eq. (47), we assume that there should exist a possibly wider family of conserved charges, including the $Q_n$'s whose first few members are written in (6), such that the corresponding one-soliton values $h_n(\eta, \epsilon)$ *form a complete basis for the space of functions of $\eta, \epsilon$.* Then, an arbitrary choice of the corresponding Lagrange parameters gives rise to an arbitrary choice of $w(\eta, \epsilon)$ as a function of both arguments. In order to make this statement precise, one would have to specify the space of functions of $\eta$ and $\epsilon$ more accurately, and possibly to construct explicitly the required, extra conserved quantities; these may be expected to be quasi-local, following the notion developed in quantum many-body systems [102]. We leave this for future research, and simply assume $w(\eta, \epsilon)$ to be an arbitrary function of both arguments.

For large values of $N$, the constraint can be expressed as bounds for the integration of the impact parameters $x_i^-$ through the TBA approach discussed in the previous section[4]

$$
\int_{\mathbb{R}^N} \prod_{i=1}^{N} dx_i^- \chi\left(u_N(x, t=0) < \varepsilon_x, x \notin [0, L]\right) \approx \prod_{i=1}^{N} \left(\int_{x_i^{\text{left}}}^{x_i^{\text{right}}} dx^-\right)
$$
$$
= \prod_{i=M+1}^{N} L_i^{\text{r}} \prod_{i=1}^{M} L_i^{\text{l}} \tag{51}
$$
$$
= L^N \prod_{i=M+1}^{N} \mathcal{K}_{N,M}^{\text{r}}(\eta_i) \prod_{i=1}^{M} \mathcal{K}_{N,M}^{\text{l}}(\eta_i).
$$

In a way, the TBA provides the Jacobian of the transformation from expression (46), in terms of the field, to expression (48), in terms of (asymptotically) free quasi-particles. Noting that we can estimate the prefactor through Sterling's formula for large values of $N$ and $M$

$$
L^N \frac{M!(N-M)!}{(N!)^2} \approx \exp\left\{N + N\log\left[\gamma^\gamma(1-\gamma)^{1-\gamma}\right] - N\log\varkappa\right\}, \tag{52}
$$

we may write the partition function (48) as

$$
\mathcal{Z}_N = \sum_{M=0}^{N-1} \int_{\Gamma_{\text{l}}^M \times \Gamma_{\text{r}}^{N-M}} \prod_{i=1}^{N} d\eta_i
$$
$$
\times \exp\left\{-\sum_{i=1}^{M}\left[w_{\text{l}}(\eta_i) - 1 + \log\varkappa - \log\left[\gamma^\gamma(1-\gamma)^{1-\gamma}\right] - \log\frac{\eta_i}{2\pi} - \log\mathcal{K}_{N,M}^{\text{l}}(\eta_i)\right] \right. \tag{53}
$$
$$
\left. - \sum_{i=M+1}^{N}\left[w_{\text{r}}(\eta_i) - 1 + \log\varkappa - \log\left[\gamma^\gamma(1-\gamma)^{1-\gamma}\right] - \log\frac{\eta_i}{2\pi} - \log\mathcal{K}_{N,M}^{\text{r}}(\eta_i)\right]\right\},
$$

where we decomposed the Gibbs weight into left- and right-contributions $w_{\text{l}}(\eta) = w(\eta, -1)$, $w_{\text{r}}(\eta) = w(\eta, +1)$.

## 3.4 Large deviation principle and Yang-Yang equation

In the thermodynamic limit $N \to \infty$, we can evaluate the partition function through the large deviation principle

$$
\mathcal{Z}_N \asymp \exp\left(-N\mathcal{F}^{\text{MF}}[\rho_{\text{l}}^*(\eta), \rho_{\text{r}}^*(\eta)]\right), \tag{54}
$$

---

[4]This essentially amounts to assuming that varying the impact parameter of a single soliton has only negligible effects on the overall gas.

where $\rho_l^*$ and $\rho_r^*$ is the (assumed unique) couple of minimizers of the mean-field free energy functional

$$
\begin{aligned}
\mathcal{F}^{\mathrm{MF}}[\rho_l(\eta), \rho_r(\eta)] = &\int_{\Gamma_l} d\eta \rho_l(\eta) \left[ w_l(\eta) - 1 + v - \log \frac{\eta \mathcal{K}_l(\eta)}{2\pi} + \log \rho_l(\eta) \right] \\
+ &\int_{\Gamma_r} d\eta \rho_r(\eta) \left[ w_r(\eta) - 1 + v - \log \frac{\eta \mathcal{K}_r(\eta)}{2\pi} + \log \rho_r(\eta) \right].
\end{aligned}
\tag{55}
$$

This amounts to applying Laplace method (or the saddle-point approximation) on Eq. (53), in which we introduced $v = \log\left[\gamma^\gamma (1-\gamma)^{1-\gamma}\right]$, and where the configuration entropy term $\log \rho$ comes from Sanov's theorem in regards to the empirical density [103]. Recalling that $v(\eta) = \pm\sqrt{1-\eta^2}$ is the momentum of the quasi-particle and introducing the right/left-occupation functions

$$
n_r(\eta) = \frac{2\pi}{\eta} \frac{\rho_r(\eta)}{\mathcal{K}_r(\eta)}, \qquad n_l(\eta) = \frac{2\pi}{\eta} \frac{\rho_l(\eta)}{\mathcal{K}_l(\eta)},
\tag{56}
$$

as well as the pseudo-energies

$$
\varepsilon_r = -\log n_r, \qquad \varepsilon_l = -\log n_l,
\tag{57}
$$

minimising the mean-field free energy functional (55) amounts to solving the system

$$
\begin{cases}
0 = \dfrac{\delta \mathcal{F}^{\mathrm{MF}}[\rho_l, \rho_r]}{\delta \rho_l}, \\[2mm]
0 = \dfrac{\delta \mathcal{F}^{\mathrm{MF}}[\rho_l, \rho_r]}{\delta \rho_r},
\end{cases}
\tag{58}
$$

or, more explicitly, one must solve a system of Yang-Yang type equations

$$
\begin{cases}
\varepsilon_l(\eta) = w_l(\eta) + v - \displaystyle\int_{\Gamma_l} \frac{d\mu}{2\pi} \varphi_O(\eta, \mu) e^{-\varepsilon_l(\mu)} - \int_{\Gamma_r} \frac{d\mu}{2\pi} \varphi_H(\eta, \mu) e^{-\varepsilon_r(\mu)}, \\[3mm]
\varepsilon_r(\eta) = w_r(\eta) + v - \displaystyle\int_{\Gamma_r} \frac{d\mu}{2\pi} \varphi_O(\eta, \mu) e^{-\varepsilon_r(\mu)} - \int_{\Gamma_l} \frac{d\mu}{2\pi} \varphi_H(\eta, \mu) e^{-\varepsilon_l(\mu)}.
\end{cases}
\tag{59}
$$

This generalises to bidirectional equations (involving two different soliton types) the Yang-Yang type equation introduced in [34] for the KdV soliton gas. These equations define a one to one map between the set of inverse temperatures $\{\beta_n\}_{n=1}^\infty$ defining the GGE and the occupation functions, $n_l$ and $n_r$, which can then be related to the DOS's, $\rho_l$ and $\rho_r$, through the NDR's (39) and the definitions (56).

## 3.5 Thermodynamic quantities

Using the previous relations (59) in Eq. (55), along with the dispersion relations (39), we may define the free energy density $\mathcal{F}$ as the minimised mean-field free energy functional

$$
\mathcal{F} = -\int_{\Gamma_l} \frac{dP(\mu)}{2\pi} n_l(\mu) - \int_{\Gamma_r} \frac{dP(\mu)}{2\pi} n_r(\mu),
\tag{60}
$$

or, using the equivalence (41)

$$
\mathcal{F} = -\int_{\Gamma_l} \frac{dP(\mu)}{\sigma_l(\mu)} - \int_{\Gamma_r} \frac{dP(\mu)}{\sigma_r(\mu)},
\tag{61}
$$

which is a straightforward generalisation to two quasi-particle types of the KdV free energy found in [34]. This then provides a straightforward way to compute the gas' entropy density $\mathcal{S}$ using the usual definition from statistical mechanics

$$\mathcal{F} = \mathcal{W} - \mathcal{S}, \tag{62}$$

where $\mathcal{W} = \int_{\Gamma_l} \mathrm{d}\eta\, w_l(\eta)\rho_l(\eta) + \int_{\Gamma_r} \mathrm{d}\eta\, w_r(\eta)\rho_l(\eta)$. Using both the Yang-Yang system (59) to eliminate both $w_l$ and $w_r$, and again the dispersion relations (39) to simplify the expression, we end up with

$$\mathcal{S} = \int_{\Gamma_l} \mathrm{d}\eta\, \rho_l(\eta)[1 - \log n_l(\eta) - \nu] + \int_{\Gamma_r} \mathrm{d}\eta\, \rho_r(\eta)[1 - \log n_r(\eta) - \nu]. \tag{63}$$

Note that at condensation $\mathcal{K}_r = \mathcal{K}_l = 0$, so that $n_r \to \infty$ and $n_l \to \infty$, meaning that $\mathcal{S}$ is indeed minimal.

## 4 Generalised hydrodynamics

### 4.1 Dressing operation

Going back to the Gibbs weights (50), we can be more specific and define it in terms of the conserved quantities carried by a single soliton (19) and some Lagrange parameters $\{\beta_n\}_{n=1}^{\infty}$

$$w(\eta) = \sum_{n=1}^{\infty} \beta_n h_n(\eta, \epsilon), \tag{64}$$

where $h_n(\eta, -1) = h_n^l(\eta)$ if the soliton is left-moving and $h_n(\eta, +1) = h_n^r(\eta)$ if it is right-moving. With this definition we can now ask ourselves how the pseudo-energies vary with respect to one of those Lagrange parameters. Given Yang-Yang equations (59) we expect

$$\begin{cases} \partial_{\beta_n} \varepsilon_l(\eta) = h_n^l(\eta) + \displaystyle\int_{\Gamma_l} \frac{\mathrm{d}\mu}{2\pi}\, \varphi_O(\eta,\mu)\partial_{\beta_n}\varepsilon_l(\mu)n_l(\mu) + \int_{\Gamma_r} \frac{\mathrm{d}\mu}{2\pi}\varphi_H(\eta,\mu)\partial_{\beta_n}\varepsilon_r(\mu)n_r(\mu), \\[2ex] \partial_{\beta_n} \varepsilon_r(\eta) = h_n^r(\eta) + \displaystyle\int_{\Gamma_r} \frac{\mathrm{d}\mu}{2\pi}\, \varphi_O(\eta,\mu)\partial_{\beta_n}\varepsilon_r(\mu)n_r(\mu) + \int_{\Gamma_l} \frac{\mathrm{d}\mu}{2\pi}\varphi_H(\eta,\mu)\partial_{\beta_n}\varepsilon_l(\mu)n_l(\mu), \end{cases} \tag{65}$$

recalling that $n_l = e^{-\varepsilon_l}$ and $n_r = e^{-\varepsilon_r}$. By writing $\partial_{\beta_n}\varepsilon_l(\eta) \equiv (h_n^l)^{l,\mathrm{dr}}(\eta)$ and $\partial_{\beta_n}\varepsilon_r(\eta) \equiv (h_n^r)^{r,\mathrm{dr}}(\eta)$ we define the left- and right- dressing operation

$$\begin{cases} h^{l,\mathrm{dr}}(\eta) = h^l(\eta) + \displaystyle\int_{\Gamma_l} \frac{\mathrm{d}\mu}{2\pi}\, \varphi_O(\eta,\mu)n_l(\mu)h^{l,\mathrm{dr}}(\mu) + \int_{\Gamma_r} \frac{\mathrm{d}\mu}{2\pi}\varphi_H(\eta,\mu)n_r(\mu)h^{r,\mathrm{dr}}(\mu), \\[2ex] h^{r,\mathrm{dr}}(\eta) = h^r(\eta) + \displaystyle\int_{\Gamma_r} \frac{\mathrm{d}\mu}{2\pi}\, \varphi_O(\eta,\mu)n_r(\mu)h^{r,\mathrm{dr}}(\mu) + \int_{\Gamma_l} \frac{\mathrm{d}\mu}{2\pi}\varphi_H(\eta,\mu)n_l(\mu)h^{l,\mathrm{dr}}(\mu), \end{cases} \tag{66}$$

so that the NDR's (39) can be reinterpreted as a dressing relation. Indeed we may define

$$\begin{cases} \rho_{s,l}(\eta) \equiv \eta\mathcal{K}_l(\eta) = \sigma_l(\eta)\rho_l(\eta) = \left|P'(\eta,-1)\right|^{l,\mathrm{dr}}(\eta) = (\eta)^{l,\mathrm{dr}}(\eta), \\[1ex] \rho_{s,r}(\eta) \equiv \eta\mathcal{K}_r(\eta) = \sigma_r(\eta)\rho_r(\eta) = \left|P'(\eta,1)\right|^{r,\mathrm{dr}}(\eta) = (\eta)^{r,\mathrm{dr}}(\eta), \end{cases} \tag{67}$$

by recalling the relations (56).

In view of the remark made after Eq. (50), we note how we assume here that $h_n(\eta, \epsilon)$ may be chosen as an arbitrary function of $\eta, \epsilon$: as is usual in many-body integrability and especially in GHD [68], the general form of the dressing operation is obtained by perturbing the "source term" $w(\eta, \epsilon)$ by an arbitrary function, thus assuming that we have access to this large space.

## 4.2 Effective velocity

Recall the nonlinear dispersion relation (67) characterising the DOS in terms of the dressing operation. Similarly it is possible to define a left- and right- flux density $f_\cdot(\eta)$. In GHD, this flux density is generally defined in terms of the dressing of the energy function $E(\eta)$

$$
\begin{cases}
\sigma_l(\eta)f_l(\eta) \equiv -\left[E'(\eta)\right]^{l,dr}(\eta), \\
\sigma_r(\eta)f_r(\eta) \equiv \left[E'(\eta)\right]^{r,dr}(\eta).
\end{cases}
\tag{68}
$$

Given our chosen convention (45) regarding the momentum function, the corresponding energy function is defined in terms of the bare velocity of the quasi-particle

$$
v(\eta,\epsilon) = \frac{E'(\eta)}{P'(\eta,\epsilon)} \quad \Rightarrow \quad E = -\frac{1}{3}\left[1-\eta^2\right]^{3/2},
\tag{69}
$$

the usual expression of the group velocity.[5] This identification of the flux density with a dressed quantity provides the second set of NDR's for our soliton gas

$$
\begin{cases}
\sigma_l(\eta)f_l(\eta) = -\eta\sqrt{1-\eta^2} + \displaystyle\int_{\Gamma_l} d\mu\, \varphi_O(\eta,\mu)f_l(\mu) + \int_{\Gamma_r} d\mu\, \varphi_H(\eta,\mu)f_r(\mu), \\
\sigma_r(\eta)f_r(\eta) = \eta\sqrt{1-\eta^2} + \displaystyle\int_{\Gamma_r} d\mu\, \varphi_O(\eta,\mu)f_r(\mu) + \int_{\Gamma_l} d\mu\, \varphi_H(\eta,\mu)f_l(\mu),
\end{cases}
\tag{70}
$$

using the definition of the dressing operation (66) along with the fact that, from Eqs. (41) and (56), $\sigma_l = 2\pi/n_l$ and $\sigma_r = 2\pi/n_r$. From this we may introduce the left- and right- effective velocity,

$$
v_l^{eff} \equiv \frac{f_l}{\rho_l}, \quad \text{and} \quad v_r^{eff} \equiv \frac{f_r}{\rho_r},
\tag{71}
$$

of solitons moving through the gas. These effective velocities can be computed by multiplying the left- and right- NDR equations (39) respectively by $v_l^{eff}$ and $v_r^{eff}$, then subtracting the second NDR relations (70), identifying both $f_l = \rho_l v_l^{eff}$ and $f_r = \rho_r v_r^{eff}$ and, ultimately, dividing by $\eta$. This yields

$$
\begin{cases}
v_l^{eff}(\eta) = v(\eta,-1) - \dfrac{1}{\eta}\left[\displaystyle\int_{\Gamma_l} d\mu\, \varphi_O(\eta,\mu)\rho_l(\mu)[v_l^{eff}(\eta)-v_l^{eff}(\mu)] + \int_{\Gamma_r} d\mu\, \varphi_H(\eta,\mu)\rho_r(\mu)[v_l^{eff}(\eta)-v_r^{eff}(\mu)]\right], \\
v_r^{eff}(\eta) = v(\eta,+1) - \dfrac{1}{\eta}\left[\displaystyle\int_{\Gamma_r} d\mu\, \varphi_O(\eta,\mu)\rho_r(\mu)[v_r^{eff}(\eta)-v_r^{eff}(\mu)] + \int_{\Gamma_l} d\mu\, \varphi_H(\eta,\mu)\rho_l(\mu)[v_r^{eff}(\eta)-v_l^{eff}(\mu)]\right],
\end{cases}
\tag{72}
$$

recovering the results presented in [30] in which the authors phenomenologically proposed a general expression for the effective velocity of a bidirectional soliton gas.

On top of that, it is well-known that the Boussinesq equation can be reduced to the KdV equation in the long-wave (or equivalently small $\eta$) limit. As an additional check, Eqs. (72) should be consistent with the effective velocity of the KdV soliton gas. Indeed, in the small $\eta$ limit the head-on an overtaking phase shifts simplify in such fashion

$$
\varphi_H \approx 0, \qquad \varphi_O \approx 2\log\left|\frac{\eta-\mu}{\eta+\mu}\right|.
\tag{73}
$$

---

[5]Note that the energy function is bounded from above and below, $-1/3 < E < 0$. Since the relevant physical quantities only depend on the derivative of the energy function, it can be offset by 1/3, to make the interpretation in terms of the energy of a quasi-particle excitation, which is expected to be positive, easier.

Left- and right-moving solitons essentially decouple, while the overtaking phase shift reduces to the KdV phase shift. Similarly the velocity of solitons simplifies to

$$v(\eta, \epsilon) \approx \epsilon \left[ 1 - \frac{\eta^2}{2} \right]. \tag{74}$$

Because of Galilean invariance, the constant part of the velocity can be absorbed through a boost and, similarly, time-reversal takes care of the factor $\epsilon$. Finally, choosing a different normalisation for the amplitude of the wave field and for the space and time variables

$$u \rightarrow 8u, \qquad x \rightarrow \sqrt{2}x, \qquad t \rightarrow \sqrt{2}t, \tag{75}$$

yields

$$\frac{\eta^2}{2} \rightarrow 4\eta^2, \quad \text{and} \quad \Delta(\eta, \mu) \rightarrow \frac{1}{\eta} \log \left| \frac{\eta - \mu}{\eta + \mu} \right|. \tag{76}$$

Putting all of this together, equations (72) eventually reduce to the single equation

$$v^{\text{eff}}(\eta) = 4\eta^2 - \frac{1}{\eta} \int_{\Gamma} \mathrm{d}\mu \log \left| \frac{\eta - \mu}{\eta + \mu} \right| \rho(\mu) \left[ v^{\text{eff}}(\eta) - v^{\text{eff}}(\mu) \right], \tag{77}$$

which is precisely the one satisfied by the KdV effective velocity first derived in [1].

## 4.3 Thermodynamic averages and correlation functions

Putting together the statistical mechanics picture we developed in section 3 with the notion of dressing introduced in section 4.1, we can now borrow general results from [43,45] pertaining to thermodynamic quantities in GHD.

What we present in this section can be seen as a straightforward generalisation of usual results from statistical mechanics. Provided our soliton gas is accurately described by the GGE defined by (48), associated with the free energy density (60), we can conjecture expressions of thermodynamic averages and correlations in our soliton gas. Indeed, given time-conserved Poisson commuting charges of type (e.g. Eqs. (6) )

$$Q_n = \int_{\mathbb{R}} \mathrm{d}x \; q_n(x), \tag{78}$$

and the associated space conserved currents

$$J_n = \int_{\mathbb{R}} \mathrm{d}x \; j_n(x), \tag{79}$$

we can easily write the ensemble averages of the densities $q_n$ and $j_n$. As per standard statistical mechanics we have

$$\begin{aligned}
\langle q_i \rangle &= \frac{\partial \mathcal{F}}{\partial \beta_n} \\
&= \int_{\Gamma_l} \frac{\mathrm{d}P(\mu)}{2\pi} n_l(\mu) h^{l,\text{dr}}(\mu) + \int_{\Gamma_r} \frac{\mathrm{d}P(\mu)}{2\pi} n_r(\mu) h^{r,\text{dr}}(\mu) \\
&= \int_{\Gamma_l} \mathrm{d}\mu \; \rho_l(\mu) h^l(\mu) + \int_{\Gamma_r} \mathrm{d}\mu \; \rho_r(\mu) h^r(\mu),
\end{aligned} \tag{80}$$

where the relation between the first and second lines comes from using the definition the dressing operation (65)-(66), along identity (57), in the definition of the free energy density (60). The third line, which is more natural, is obtained thanks to the symmetry of the dressing operation [68]

$$
\sum_{a=\text{l,r}} \int_{\Gamma_a} \mathrm{d}\mu \; g^a(\mu) n_a(\mu) h^{a,\text{dr}}(\mu) = \sum_{a=\text{l,r}} \int_{\Gamma_a} \mathrm{d}\mu \; g^{a,\text{dr}}(\mu) n_a(\mu) h^a(\mu), \tag{81}
$$

while recalling the NDR's (67) and definitions (56). Similarly, space-integrated connected correlations are obtained as

$$
\begin{aligned}
\mathsf{C}_{ab} &\equiv \int \mathrm{d}x \; (\langle q_a(x) q_b(0)\rangle - \langle q_a(x)\rangle \langle q_b(0)\rangle) = -\frac{\partial^2 \mathcal{F}}{\partial \beta_a \partial \beta_b} \\
&= \int_{\Gamma_\text{l}} \mathrm{d}\eta \, \rho_\text{l}(\eta) h_a^{\text{l,dr}}(\eta) h_b^{\text{l,dr}}(\eta) + \int_{\Gamma_\text{r}} \mathrm{d}\eta \, \rho_\text{r}(\eta) h_a^{\text{r,dr}}(\eta) h_b^{\text{r,dr}}(\eta).
\end{aligned} \tag{82}
$$

The averages of currents can then be obtained by differentiating the free energy flux introduced in [4]

$$
\mathcal{G} = -\int_{\Gamma_\text{l}} \frac{\mathrm{d}E(\mu)}{2\pi} \, n_\text{l}(\mu) - \int_{\Gamma_\text{r}} \frac{\mathrm{d}E(\mu)}{2\pi} \, n_\text{r}(\mu). \tag{83}
$$

This expression is similar to that of the free energy density (60) in which the momentum measure has been replaced by the energy measure. It yields fairly natural results regarding the average of the currents

$$
\begin{aligned}
\langle j_i \rangle &= \frac{\partial \mathcal{G}}{\partial \beta_n} \\
&= \int_{\Gamma_\text{l}} \frac{\mathrm{d}E(\mu)}{2\pi} n_l(\mu) h^{\text{l,dr}}(\mu) + \int_{\Gamma_\text{r}} \frac{\mathrm{d}E(\mu)}{2\pi} n_r(\mu) h^{\text{r,dr}}(\mu) \\
&= \int_{\Gamma_\text{l}} \mathrm{d}\mu \, v_\text{l}^{\text{eff}}(\mu) \rho_\text{l}(\mu) h^\text{l}(\mu) + \int_{\Gamma_\text{r}} \mathrm{d}\mu \, v_\text{r}^{\text{eff}}(\mu) \rho_\text{r}(\mu) h^\text{r}(\mu),
\end{aligned} \tag{84}
$$

and the field-current correlator

$$
\begin{aligned}
\mathsf{B}_{ab} &\equiv \int \mathrm{d}x \; (\langle q_a(x) j_b(0)\rangle - \langle q_a(x)\rangle \langle j_b(0)\rangle) = -\frac{\partial^2 \mathcal{G}}{\partial \beta_a \partial \beta_b} \\
&= \int_{\Gamma_\text{l}} \mathrm{d}\eta \, v_\text{l}^{\text{eff}}(\eta) \rho_\text{l}(\eta) h_a^{\text{l,dr}}(\eta) h_b^{\text{l,dr}}(\eta) + \int_{\Gamma_\text{r}} \mathrm{d}\eta \, v_\text{r}^{\text{eff}}(\eta) \rho_\text{r}(\eta) h_a^{\text{r,dr}}(\eta) h_b^{\text{r,dr}}(\eta).
\end{aligned} \tag{85}
$$

Finally, the Drude weight characterising ballistic transport can be written, via the Kubo formula, in terms of a time integral of correlations

$$
\begin{aligned}
\mathsf{D}_{ab} &\equiv \lim_{t\to\infty} \frac{1}{2t} \int_{-t}^{t} \mathrm{d}\tau \int \mathrm{d}x \; (\langle j_a(x,\tau) j_b(0,0)\rangle - \langle j_a(x,\tau)\rangle \langle j_b(0,0)\rangle) \\
&= \int_{\Gamma_\text{l}} \mathrm{d}\eta \left( v_\text{l}^{\text{eff}}(\eta)\right)^2 \rho_\text{l}(\eta) h_a^{\text{l,dr}}(\eta) h_b^{\text{l,dr}}(\eta) + \int_{\Gamma_\text{r}} \mathrm{d}\eta \left( v_\text{r}^{\text{eff}}(\eta)\right)^2 \rho_\text{r}(\eta) h_a^{\text{r,dr}}(\eta) h_b^{\text{r,dr}}(\eta).
\end{aligned} \tag{86}
$$

## 4.4 Euler hydrodynamics

We shall now consider a weakly non-homogeneous, out of equilibrium gas in which space-time variations of the DOS's and of the effective velocities occur over macroscopic scales that

are much larger than the typical width of a soliton. In that case we may, as is standard in hydrodynamic theories [38], assume local entropy maximisation and propagation of such local equilibria.

The main ingredient in our hydrodynamic construction is the infinite set of local conservation laws

$$\partial_t q_n + \partial_x j_n = 0, \tag{87}$$

induced by the integrability structure of the Boussinesq equation. As an illustration, recalling equations (6), we present here the first three charge and current densities

$$
\begin{aligned}
q_1 &= u, & j_1 &= w_x, \\
q_2 &= w_x, & j_2 &= -u + 6u^2 + u_{xx}, \\
q_3 &= u w_x, & j_3 &= \frac{(w_x)^2 + (u_x)^2}{2} - \frac{u^2}{2} - 4u^3 - u u_{xx}.
\end{aligned}
\tag{88}
$$

We then assume our system can be divided into mesoscopic, thermodynamically large fluid cells, that are still small compared to the scales associated with the variations of the statistical properties of the soliton gas. In other words, we work under the hydrodynamic approximation: we assume that the averages of local observables can be well approximated, at large times, by averages evaluated in local GGEs

$$\langle o(x,t) \rangle \approx \langle o \rangle_{w(\eta,\epsilon;x,t)} \equiv \bar{o}_n(x,t). \tag{89}$$

We may now write mesoscopic conservation equations by averaging the microscopic laws (87) over such a fluid cell[6]

$$\partial_t \bar{q}_n(x,t) + \partial_x \bar{j}_n(x,t) = 0. \tag{90}$$

Since $\bar{q}_n(x,t)$ and $\bar{j}_n(x,t)$ are related to the DOS's and effective velocities through equations (80)-(84), provided the set of $\{h_n\}$'s is complete (which should be the case since the scattering of solitons is 2-body reducible [3]) one eventually obtains the fundamental equation of (Euler scale) GHD

$$\partial_t \rho.(\eta;x,t) + \partial_x \left[ v^{\text{eff}}.(\eta;x,t) \rho.(\eta;x,t) \right] = 0. \tag{91}$$

Most of the interesting phenomenology of the GHD of the Boussinesq equation comes from the fact that the scattering shift can be either positive or negative, leading to what can be interpreted as positive or negative "refraction" of solitons, as well as complicated interference patterns. Examples of this are discussed in more details in [90], in which simulations of so-called "polychromatic" gases (generated from exact $N-$soliton solutions with randomly distributed spectral and impact parameters) are presented and compared to exact solutions of Eqs. (91)-(72).

## 5 Conclusion

In this paper, we have constructed the generalised hydrodynamics of the soliton gas associated with the Boussinesq integrable PDE, a theory for a field lying in one dimension of space and one of time. We have constructed its hydrodynamics – the kinetic equation – and thermodynamics – the free energy. We have done this by simply following the general intuitive rules from soliton gases [2] and generalised hydrodynamics [4,5] according to which both the

---

[6]Formally we integrate equations (87) over a contour $[0,X] \times [0,T]$, make the substitutions $\frac{1}{T}\int_0^T q_n \mathrm{d}t \approx \bar{q}_n$ and $\frac{1}{X}\int_0^X q_n \mathrm{d}t \approx \bar{q}_n$ according to the law of large numbers, and end up with equations (90) provided $\bar{q}_n$ and $\bar{j}_n$ are differentiable.

hydrodynamics and thermodynamics can be obtained from the known two-soliton scattering shifts. This follows, and in some sense generalises, the analysis done for the soliton gas of the KdV equation in [34] to bidirectional models. In particular, it was important to correctly account for the choice of the "momentum function" in order to construct the thermodynamics. By contrast to the KdV case, in the Boussinesq case the most natural momentum function, with respect to the known soliton constructions, is not the physical momentum.

We note that the soliton phenomenology of the Boussinesq equation is significantly more complex than that of the KdV equation. But in some regimes of parameters, on which we have concentrated, it is nevertheless well understood, and, interestingly, includes both head-on and overtaking interactions. In certain cases, non-soliton-conserving, yet integrable, interactions can occur, but, in the chosen regime of parameters, they cannot occur in macroscopic number, hence to not affect the hydrodynamics or thermodynamics. A natural extension of the present work would be to analyse other regimes of parameters, where it may be possible to include a macroscopic number of non-soliton-conserving interactions. However, under different regimes of parameters, the $N$—soliton solution is known to develop singularities in finite time [101] that may only remain for a finite duration. One would then need to develop a better understanding of the physical meaning of such singularities, as well as carefully study their impact on the thermodynamics of the gas. Alternatively, one may have to work with non-zero background, which seems to make the phenomenology even richer as observed in [104], a priori allowing for more than two resonances while keeping the solution regular at all time. Going beyond the simple Boussinesq equation, this could prove significant for GHD in general as a prototypical example of an integrable model in which the quasi-particles are unstable and in which their number is not conserved.

Traditionally, in the IST description of the Boussinesq equation [105–107], solitons may live on top of a radiation background, associated with the continuous part of the Lax spectrum, which we have not considered here. In the hydrodynamic limit, it is natural to expect this radiation background to be present in a generic fashion and, as such, our construction could be interpreted as a toy model. However there are several reasons to think that a soliton gas description might be "complete", notably the fact that, in the $N \to \infty$ limit, $N$—soliton solutions can provide a good approximation of a general solution involving radiations as well (see Chapter 3, Section 8 of [9]). And indeed, recently, soliton gas solutions with non-zero reflection coefficient (indicating the presence of radiations) were constructed rigorously as the $N \to \infty$ limit of $N$—soliton solutions for the KdV and modified KdV equations [19, 108]. Moreover, error estimates for the approximation of a general solution by $N$—soliton solutions on a compact $(x, t)$ domain were given in [109] for the NLS hierarchy. We expect those results, at least on the qualitative level, should translate to the Boussinesq equation as well.

On top of all that, it should be possible to extend the construction presented in this paper by straightforwardly adapting known GHD results. GHD, just like conventional hydrodynamics, is fundamentally a derivative expansion. While the equation (91) generalises the Euler equations to account for integrable systems, it is possible to compute an additional diffusion operator [56, 57] (or even dispersion [59]), which would transform equation (91) into the GHD equivalent of the Navier-Stokes equation. It is also possible to probe integrability breaking, either by adding an external potential [110] or through space-time inhomogeneous interactions [47], which would result in the addition of an "effective acceleration" term in equation (91). This suggests it should be possible to ultimately adapt the soliton gas approach developed in this paper to study the generalisation of the Boussinesq equation introduced in [80] as a model for neural activity.

Beyond that the primary interest of developing the soliton gas theory for the Boussinesq equation is its eventual application to the KP equation, a PDE for a field in *two* dimensions of space. Indeed, it is known, as we have recalled, that certain "essentially stationary" solutions

of the KP equation (stationary in Galilean frames with non-zero velocities) in fact are simply space-time solutions of the Boussinesq equation, where the Boussinesq time is identified with one of the KP space directions. This relation, which we study in more details in [90], will lay the ground for developing the soliton gas theory for the KP equation, and will help reveal the structure of the theory beyond models in one dimension of space.

## A  GHD of the "bad" Boussinesq soliton gas

Generalisation of the previous construction to the bad Boussinesq equation is fairly straightforward once the main building blocks – namely the scattering shifts, the momentum and energy functions, and the dressing operations – have been provided.

First of all, the bad Boussinesq equation admits $N-$soliton $\tau-$functions in the form (13) except, this time, the phase is given by

$$\theta_i(x,t) = \eta_i\left(x - \epsilon_i t\sqrt{1+\eta_i^2} - x_i^0\right),\tag{A.1}$$

and the phase shift by

$$\varphi_{ij} = \log\frac{\left(\epsilon_i\sqrt{1+\eta_i^2} - \epsilon_j\sqrt{1+\eta_j^2}\right)^2 + 3(\eta_i - \eta_j)^2}{\left(\epsilon_i\sqrt{1+\eta_i^2} - \epsilon_j\sqrt{1+\eta_j^2}\right)^2 + 3(\eta_i + \eta_j)^2}.\tag{A.2}$$

The main difference here is that the argument of the square root, in both the expression of the soliton velocity and of the phase shift, has changed from $(1-\eta^2)$ to $(1+\eta^2)$. This, along the fact that the argument of the log, in the expression of the phase shift, is now a ratio of sums of squares, and not of differences, means that the $N-$soliton solution remains regular at all time for any set of real spectral parameters. Despite the bad Boussinesq being ill-posed for generic arbitrary initial data, conditions for the regularity of its $N-$soliton solution are less restrictive than they are for the good Boussinesq equation. Additionally, the phenomenology is not as rich, the phase shift is always negative and there are no bound states.

Then, by convention we choose the same momentum function we did earlier for the good Boussinesq equation which, given the velocity of solitons, also fixes the energy function

$$P(\eta,\epsilon) = \epsilon\frac{\eta^2}{2}, \qquad E(\eta) = \frac{1}{3}\left[1+\eta^2\right]^{3/2}.\tag{A.3}$$

The dressing operation takes again the form (66), changing the phase shift accordingly, for which infer the NDR's

$$\begin{cases}\sigma_l(\eta)\rho_l(\eta) = (\eta)^{l,dr}(\eta),\\ \sigma_r(\eta)\rho_r(\eta) = (\eta)^{r,dr}(\eta),\end{cases}\quad\text{and}\quad\begin{cases}\sigma_l(\eta)f_l(\eta) = -\left(\eta\sqrt{1+\eta^2}\right)^{l,dr}(\eta),\\ \sigma_r(\eta)f_r(\eta) = \left(\eta\sqrt{1+\eta^2}\right)^{r,dr}(\eta),\end{cases}\tag{A.4}$$

which yields again an effective velocity of form (72) by taking the ratio $v_\cdot^{\text{eff}} \equiv f_\cdot/\rho_\cdot$. Generalisation of any other identity constructed for the good Boussinesq equation is then immediate.

## Acknowledgments

This work has benefited from discussions with Gino Biondini, Gennady El and Giacomo Roberti. We are especially thankful for their comments regarding early versions of this manuscript.

**Funding information**    The authors were supported by the Engineering and Physical Sciences Research Council (EPSRC) under grant EP/W010194/1.

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
