# Peer review of "Soliton gas of the integrable Boussinesq equation and its generalised hydrodynamics"

_SciPost Physics, doi:SciPost Phys. 18, 075 (2025)_

## Round 1 · Referee Report · Anonymous (Referee 1) · 2024-8-5

Strengths

  1. Timely topic
  2. State-of-the-art methods
  3. Potential extension to 2+1 dimensional physics

Weaknesses

  1. The novelty of the results and their physical relevance are not clearly articulated.
  2. No example results given
  3. Notation is confusing in some places
  4. Text needs thorough grammatical and stylistic revision

Report

This paper considers the generalised hydrodynamics of the integrable Boussinesq equation. This is an interesting problem as it paves the way to a 2+1-dimensional extension of the GHD. The scientific relevance of the results justifies publication; however, the level of the results justifying publication in Scipost Physics is not established. In particular, it is unclear to what extent the work goes beyond a straightforward exercise of applying the GHD to another system. The paper would greatly benefit from some examples of time evolution, showing physical features of the dynamics, or some analysis of particular physical properties of the model. In the present form, I propose transferring the paper to Scipost Physics Core.

Requested changes

  1. I found the notations a little confusing in connection with the good/bad Boussinesq equation distinction. In (4), the notation omega, which is normally used for angular frequency, is instead used for the imaginary units times the angular frequency. While notations are indeed arbitrary, they definitely made reading harder, all the time having to remind myself of this convention, which is not even pointed out in the text and which goes contrary to the usual one used in the physics literature.

Additionally, I could only assume that the “large deviation principle” is essentially the same as “saddle point approximation”, at least (33), and the subsequent reasoning certainly indicates so. I propose that the authors make this connection clear, as it would make the paper much more readable for a large part of the audience.

I also find the negative sign of the energy function as given in eqn. (68) strange, as this is hard to interpret as the energy of some quasi-particle excitation in the system, which is expected to be positive to keep the energy bounded from below. While this may be a consistent convention, some clarification would be helpful at this point.

  1. The use of tenses is confusing and goes against the usual ones in research papers. For example, the authors use future tense “we shall see”, “we shall develop”, etc., referring to other parts of the paper. Other examples are “we will now drop”, “it will be instead more convenient”, and “we will assume”. In all these places, a simple present is the appropriate one to use. Also, present perfect is appropriate when reviewing previous literature, but in places like “we have considered simultaneously the overtaking…”, or the conclusions, a simple past is appropriate.

There are also typos, such as e.g. “litteral” instead of literal; running a spell-checker should greatly help.

  1. Section 4.4 on the Euler hydrodynamics is essentially trivial as it repeats the well-known reasoning to exchange the conservation laws for density equations in GHD. It would make much more sense to present some example solutions of (90), pointing out some characteristic behaviour, answering questions such as
  2. Is there any physical novelty compared to other applications of the GHD?
  3. If yes, how are these connected with some special structure of the Boussinesq equation?

Recommendation

Accept in alternative Journal (see Report)

---

## Round 1 · Referee Report · Anonymous (Referee 2) · 2024-8-13

Strengths

The paper provides a detailed study of the soliton gas governed by the integrable Businesq equation. While ingredients for such enterprise are known, the issue how to implement them is a difficult task. Interesting novel points are to exploit directly the multi-soliton solution of Hirota and the observation to rewrite the TBA equation as a two component system. The text is very well-written. Sufficient background is provided so to follow the deductions

Weaknesses

For other integrable models, beyond the soliton gas, there is also a "radiation" background. In the present case is there such a contribution? If yes, why can it be ignored in a hydrodynamic description?

Report

The submitted paper satisfies the level of SciPost Physics. Publication is recommended, subject to the point raised above.

Recommendation

Publish (easily meets expectations and criteria for this Journal; among top 50%)

---

## Round 1 · Referee Report · Anonymous (Referee 3) · 2024-9-8

Strengths

  1. Provides a novel and synergetic link between generalized hydrodynamics and soliton gas theories for a canonical bidirectional nonlinear wave model.

  2. Opens a pathway for the construction of two-dimensional soliton gas theory via the connection between the Boussinesq and the Kadomtsev-Petviashvili equations.

Weaknesses

  1. Uses some phenomenological assumptions that are not rigorously proved (e.g. soliton resolution of a random, rapidly decaying Boussinesq wave field in the "time-of-flight" construction).

Report

The manuscript is concerned with the construction of generalized hydrodynamics (GHD) of soliton gas for the integrable “two-way” Boussinesq equation. Specifically, the authors extend the previous construction of unidirectional soliton gas GHD for the KdV equation to the (bidirectional) Boussinesq equation. This is an important extension in several respects. The main motivation is the connection of the Boussinesq equation with the Kadomtsev-Petviashvili (KP) equation, a universal integrable model for weakly two-dimensional nonlinear waves. Namely, the Boussinesq equation represents a stationary (2+0) reduction of the KP equation so the GHD of the Boussinesq soliton gas could provide important insights into the behavior of the general KP soliton gas. At the same time, the GHD of the Boussinesq soliton gas is important on its own as the Boussinesq equation is a canonical integrable model exhibiting anisotropic soliton interactions, with the head-on and overtaking collision phase shifts having different signs with important consequences for the macroscopic observables in the soliton gas. Generally, the phenomenology of soliton interactions in the Boussinesq equation is much richer than that for the KdV equation. As a result, the GHD generalization of the “isotropic” KdV gas to the anisotropic Boussinesq case is quite nontrivial and exhibits a number of qualitatively new features. In addition to the hydrodynamics of the Boussinesq solitons gas, the authors develop its thermodynamics (entropy, free energy, temperature, etc.) and compute correlations, not available via the standard soliton gas machinery.
The paper is written very well, offering a thorough introduction to the properties of multisoliton solutions of the Boussinesq equation and outlining important parallels between the Boussinesq soliton gas GHD (TBA, GGE, dressing operation, etc.) and the key quantities and relations of its spectral/kinetic theory (density of states, spectral scaling function, nonlinear dispersion relations etc). Overall, this paper represents a valuable contribution to the rapidly growing literature on soliton gas that will have impact on two major communities in modern mathematical and theoretical physics.

I believe the paper meets the SciPost publication criteria and have no hesitation recommending its publication in the essentially present form subject to minor edits suggested below.

Requested changes

1. P.5. Second line after Eq. 5: “Indeed, one can see by direct substitution that the Lax equation is equivalent to the original equation (1)”. This should be augmented by the isospectrality condition, e.g. “provided the spectrum of the operator $L$ is time-invariant”.

2. P. 5, the line before Eq. (6), replace “conserved charges” by “conserved charges (densities)”.

3. After Eq. (10), replace $|t|=\infty$ and $|x| = \infty$ with $|t| \to \infty$ and $|x| \to \infty$.

4. The line before Eq. (14): “phase shift” should be “phase shifts”.

5. P.6, the last sentence. I suggest presenting explicit expressions for the KdV phase shifts or simply making a reference to the relevant expressions in Section 4.2

6. P.9 section 2.6: “resonnant” should be “resonant”.

7. P.10. Section 3, the introductory paragraph. “litteral” should be “literal”.

8. P.12, 2 lines before Eq. (40); “Equations (38) is akin…” should be “equations (38) are akin…”.

9. Make a reference to Ref. [31] when introducing the spectral scaling functions after Eq. (40)

10. P.12. Introduce the acronym NDR’s here as it is used later in the text.

Recommendation

Publish (easily meets expectations and criteria for this Journal; among top 50%)

---

## Round 2 · Referee Report · Anonymous (Referee 3) · 2024-12-12

Report

I am happy with the revised version of the manuscript and recommend publication in the present form.

Recommendation

Publish (easily meets expectations and criteria for this Journal; among top 50%)

---

## Round 2 · Referee Report · Anonymous (Referee 2) · 2024-12-14

Report

My comments on the previous version have been very satisfactorily
addressed. The submitted paper reports on novel and interesting
advances on constructiing GHD for pure soliton gases.
I recommend publication as it is. In my opinion this is a very strong article.

Recommendation

Publish (surpasses expectations and criteria for this Journal; among top 10%)

---

## Round 2 · Author Response

We thank the referees for the careful reading of the manuscript and the useful comments which helped improve the presentation and make it clearer. We furthermore apologise for the delayed reply, while an updated version of the manuscript was put on arXiv soon after having received the editor's recommendation, both the authors got busy and the final steps of the resubmission were put on standby.

Report #1 by Referee 2

  1. Notation conventions have been changed when discussing the difference between the Good and Bad Boussinesq equation.

Similarities between the large deviation principle and the saddle point approximation were highlighted.

We added a footnote page 17 to discuss the sign of the energy function. The short of it is that this convention was chosen in order to ensure that the momentum function of right-moving solitons would be positive, while that of left-moving would be negative. Furthermore, the expression of the energy function, combined with the restrictions on the spectral parameter \eta, ensures it is bounded from below.

  1. We corrected the typos, but did not modify our use of tenses.

  2. A small paragraph was added at the end of page 20 highlighting some specificities of the Boussinesq phenomenology, as well as making reference to another paper by the authors in which those specificities are discussed further. Said paper also features some examples of solutions to Eq. (90).

Report #2 by Referee 1

A paragraph was added in Section 5, at the end of page 21 to highight the relevance of a gas made of only solitons and why one may forego radiations. The main reason is that, in the large N limit, N-soliton solutions can provide excellent approximations of general solutions (even ones involving radiations) on any compact (x,t) domain (see e.g. refs 9, 19 and the newly cited 108, 109).

Report #3 by Referee 3

All the comments by Referee 3 were taken into account, c.f. the list of changes.

---

## Round 2 · List of Changes

Several typos were corrected.

Some further references were added in the introduction.

We changed the notations of Section 2.1 in accordance with the suggestions of Referre 2.

Mention of the iso-spectrality condition was added.

A footnote was added page 6 to clarify the origin of the name "spectral parameter".

We added the explicit expression for the KdV phase shift and made a reference to section 4.2 at the beginning of page 7.

A reference to [31] was added after equation (40), page 12, and the acronym NDR was introduced in the same sentence.

Mention of the link between the large deviation principle and the saddle point approximation was added page 15.

A footnote was added page 17 to discuss the sign of the energy function.

A paragraph was added at the end of page 20 highlighting some specificities of the Boussinesq phenomenology, as well as making reference to another paper by the authors in which those specificities are discussed further.

A paragraph was added in Section 5, at the end of page 21 to highight the relevance of a gas made of only solitons and why one may forego radiations.

---

## Editorial Decision

published